# Quantifying Cardinal Temperatures of Chia (*Salvia hispanica* L.) Using Non-Linear Regression Models

**DOI:** 10.3390/plants11091142

**Published:** 2022-04-22

**Authors:** Daniel Cabrera-Santos, Cesar A. Ordoñez-Salanueva, Salvador Sampayo-Maldonado, Jorge E. Campos, Alma Orozco-Segovia, Cesar M. Flores-Ortiz

**Affiliations:** 1Laboratorio de Fisiología Vegetal, Unidad de Biología, Tecnología y Prototipos (UBIPRO), Facultad de Estudios Superiores Iztacala, Universidad Nacional Autónoma de México, Av. de los Barrios No. 1, Los Reyes Iztacala, Tlalnepantla C.P. 54090, Mexico; danielcabsantos@comunidad.unam.mx (D.C.-S.); caos@unam.mx (C.A.O.-S.); ssampayom@hotmail.com (S.S.-M.); 2Laboratorio de Bioquímica Molecular, Facultad de Estudios Superiores Iztacala, Universidad Nacional Autónoma de México, Tlalnepantla C.P. 54090, Mexico; jcampos@unam.mx; 3Departamento de Ecología Funcional, Instituto de Ecología, Universidad Nacional Autónoma de México, Coyoacán, Mexico City C.P. 04510, Mexico; aorozco@ecologia.unam.mx; 4Laboratorio Nacional en Salud, Facultad de Estudios Superiores Iztacala, Universidad Nacional Autónoma de México. Av. de los Barrios No. 1, Los Reyes Iztacala, Tlalnepantla C.P. 54090, Mexico

**Keywords:** beta functions, cardinal temperatures, intersected-line models, *Salvia hispanica* L., segmented non-linear regressions

## Abstract

Temperature is the main factor that impacts germination and therefore the success of annual crops, such as chia (*Salvia hispanica* L.), whose seeds are known for their high nutritional value related to its oil. The effect of temperature on germination is related to cardinal-temperature concepts that describe the range of temperature over which seeds of a particular species can germinate. Therefore, in this study, in addition to calculated germinative parameters such as total germination and germination rate of *S. hispanica* seeds, the effectiveness of non-linear models for estimating the cardinal temperatures of chia seeds was also determined. We observed that germination of *S. hispanica* occurred in cold to moderate-high temperatures (10–35 °C), having an optimal range between 25 and 35 °C, with the highest GR and t_50_ at 30 °C. Temperatures higher than 35 °C significantly reduced germination. Output parameters of the different non-linear models showed that the response of chia germination to temperature was best explained by beta models (B). Cardinal temperatures calculated by the B1 model for chia germination were: 2.52 ± 6.82 °C for the base, 30.45 ± 0.32 °C for the optimum, and 48.58 ± 2.93 °C for the ceiling temperature.

## 1. Introduction

Chia (*Salvia hispanica* L.) is a summer biannual herbaceous and oleaginous plant, belonging to the family of *Lamiaceae*, rich in officinal and aromatic species with essential oils (EOs) making them valuable in many fields as cosmetics, food, medicine [1,2], and in agriculture as antimicrobial agents [3,4,5]. This oilseed crop is native to the region comprising mountainous areas from midwestern Mexico to northern Guatemala [6,7,8]. Historically, chia has been cultivated in subtropical and frost-free regions [9], specifically in the mountainous areas of the Pacific Ocean slope [10]. Currently, chia is cultivated in Australia, Bolivia, Colombia, Guatemala, Mexico, Peru, and Argentina [11].

Chia seeds vary in size from 1 to 2 mm with oval and flattened shapes, with colors from black to white, with or without gray and black spots [12,13,14,15]. Seed compositions are 15–25% protein, 30–33% fat, 26–41% carbohydrates, 18–30% fiber, 4–5% ashes, and also minerals, vitamins, and dry matter [12,16]. Seed oil represents 25–40% of the total seed weight and is composed of almost 50–57% of linolenic and 17–26% linoleic acid (ω-3 and ω-6 fatty acids, respectively), dietary fiber (over 30% of the total weight), and proteins of high biological value (around 19% of the total weight) [17,18,19,20]. So far, there is no evidence of adverse effects or allergenicity caused by whole or ground chia seeds [21]. These chia characteristics have a positive impact on nutrition and health, and since 2009 have been approved as a novel food by the European Parliament and the European Council [22].

Due to public health awareness and the demand for functional food with innumerable health benefits, chia production has experienced an increase in its production worldwide [16]. On this subject, during the year 2018, the chia seed worldwide market was valued at USD 66.5 million; by the year 2024, the market is projected to reach a value of USD 88.1 million [23]. Experimental evidence about *S. hispanica* is still in progress, especially concerning seed technology, including the role of temperature on germinative behavior [24,25,26]. Therefore, it is necessary to obtain enough knowledge about the factors underlying germination and plant development, following different approaches to support chia agricultural practices.

Germination and seedling establishment comprise two critical stages, on which the success of the next generation depends. Germination is affected by temperature, water availability, and gaseous environment [27]. From these environmental parameters, when water availability is not a limitation, temperature is the main factor controlling germination, exerting influence on germination rate, latency level, and seed deterioration rate. Germination rate is mainly affected by temperature because it is related to water absorption by seeds [28,29]. Moreover, biochemical reaction rates underlaying the metabolic networks are affected [30]. In addition, the time at which germination occurs could be affected by temperature [31].

Thermal-germination models, which usually contain certain assumptions about within-population variability in germination-rate response to temperature, are one way to characterize germination reaction to temperature [32]. These assumptions are most often related to cardinal-temperature concepts that describe the range of temperature over which seeds of a particular species can germinate. Three cardinal temperatures have been recognized: Base temperature (*T_b_*) below which germination does not proceed; an optimal temperature (*T_o_*) at which the rate of germination is highest; and a maximum or ceiling temperature (*T_c_*) above which germination ceases [33,34,35,36,37,38,39,40]. The *T_b_* for germination of any fraction of the seed population is considered to be a constant, while *T_c_* varies among each percentile fraction in a normal distribution [35,36]. The temperature has an impact on plant growth and development, so estimating the cardinal temperatures is essential.

Because germination is one of the most important factors in the success of annual crops, playing a key role in crop production, practical research in plant science usually attempts to establish the minimum temperature required for germination or its maximum range. To improve establishment success rates and to reduce costs, it is essential to have a good understanding of seed germination requirements of species of agricultural importance. While several models including linear and non-linear functions are available to estimate cardinal temperatures, a suitable model for the specific crop should be selected.

Therefore, in this study, in addition to calculated germinative parameters such as total germination and germination rate of *S. hispanica* seeds, the effectiveness of non-linear models (segmented linear regressions and beta functions) for estimating the cardinal temperatures of chia seeds was also determined. Consequently, due to its economic potential based on its positive effects on human health, such knowledge may be useful for identifying the best planting dates for this oilseed crop in a range of climates and regions, and importantly, its resistance and distribution concerning climate change scenarios.

## 2. Results

### 2.1. Germination

Cumulative germination data were transformed in Arcsine, Logit and Ln; however, data did not fulfill the normality test in any case; therefore, non-parametric tests were performed.

In the group of treatments of 10–35 °C, average germination reached >88%, while the highest final germination observed was 98% at 20 °C. High temperatures, i.e., 40 °C and 45 °C, with inhibited germination reached final germination values of 44% and 11%, respectively, the latter being the lowest value observed of the eight treatments (Figure 1 and Table 1). In accordance with these results, when the group of treatments between 10 and 35 °C were analyzed separated from 40 °C and 45 °C treatments, no significant differences were observed in the final germination percentage (H (5) = 9.671, *p =* 0.085 for Kruskal-Wallis test). However, when the latest groups were analyzed together with 40 °C and 45 °C treatments, significant differences were observed between each of the treatments from 10 to 35 °C, with regards to 40 °C and 45 °C (H (7) = 28.27, *p <* 0.001 for Kruskal-Wallis test and *p ≤* 0.026 for pairwise Dunn’s test).

Concerning the time required to reach 50% germination (t_50_), the highest value was observed at 30 °C (0.30 ± 0.10 days) and the lowest at 10 °C (5.50 ± 0.44 days), while the treatments at 40 °C and 45 °C did not reach 50% germination. t_50_ at 30 °C was 18.3-fold, 6.6-fold, 4.4-fold, 1.8-fold, and 1.2-fold faster than 10, 15, 20, 25, and 35 °C, respectively (Table 1). To reach t_50_, 1–5 days elapsed at temperatures below 20 °C, whereas, in the temperature range of 25–35 °C, less than one day was required. In the group of 10–35 °C, significant differences were observed between all treatments (H (5) = 26.11, *p <* 0.001 for Kruskal-Wallis test and *p ≤* 0.034 for pairwise Dunn’s test) (Table 1).

For GR values it was possible to distinguish three different groups: 10–20 °C, where a gradual increase in GR was observed, reaching a plateau at 25–35 °C, and finally 40–45 °C, where a decrease was observed. The highest GR was observed at 30 °C with 22 seeds per day and the lowest at 45 °C with about two seeds per day (Figure 2). Significant differences were observed between the treatments (H (7) = 36.58, *p <* 0.001 for Kruskal-Wallis test, and *p ≤* 0.020 for pairwise Dunn’s test).

### 2.2. Cardinal Temperature Determination by Linear and Non-Linear Regression Models

Table 2 summarizes the six non-linear models used for cardinal temperature determination fitted to the reciprocal of the germination time versus temperature data for each of the 10–80% percentiles for each of the treatments (10–45 °C).

For the S1 model (Figure 3, S1) the estimated average base, optimum, and ceiling temperatures were 6.90 ± 1.86 °C; 33.45 ± 2.76 °C, and 42.83 ± 3.88 °C, respectively (Table 3). For the S2 model, calculated Tb, To, and Tc were 6.65 ± 2.55 °C, 36.97 ± 5.70 °C, and 44.96 ± 1.45 °C, respectively (Figure 3, S2). In both models, it was only possible to calculate the cardinal temperatures from 10–40% percentiles, due to the few available points of 50–80% percentiles to perform the respective non-linear regressions. On the S3 model (Figure 3, S3) calculated Tb, To, and Tc were 6.52 ± 2.55 °C, 32.60 ± 1.20 °C, and 41.34 ± 3.74°C, respectively. On the B1 model, Tb, To, and Tc for 10–80% were 2.52 ± 6.82 °C, 30.45 ± 0.32 °C, and 48.58 ± 2.93 °C, respectively. On the B2 model, Tb, To, and Tc were 9.74 ± 2.23 °C, 31.24 ± 0.21 °C, and 44.10 ± 1.48 °C, respectively. Finally, on the B3 model, the values were 4.97 ± 4.06 °C, 28.44 ± 2.28 °C, and 44.26 ± 2.83 °C, for Tb, To, and Tc, respectively. The optimum temperature for all six models was very close to 30 °C, the temperature at which the higher GR and t_50_ were observed (Table 1 and Figure 2).

For segmented models, the Tb, To, and Tc values varied among percentiles in the three models (Figure 3; Table 3, Table 4 and Table 5). For the S1 model, the Tb variation difference from the lowest to the highest percentile temperature estimation was 5.94 °C, the To was 1.5 °C, and the Tc was 6.91 °C; the variation range was wider for To than Tb and Tc. In the S2 model, Tb varies in a wider range of 7.03 °C, To varies by about 18.15 °C and Tc varies in a range of 2.45 °C. For the S3 model, Tb varies in a range of 9.1 °C, To varies 4.05 °C, and Tc in a range of 11.62 °C. For the B1 model (Table 6), Tb variation difference was 17.38 °C (for 50–80% estimated Tb was negative), To difference was 0.87 °C, and Tc difference was 6.47 °C; for the B2 model Tb variation difference was 9.48 °C, To was 5.42 °C, and Tc difference was 7.38 °C; and finally, for the B3 model Tb variation difference was 12.85 °C, To was about 1.13 °C, and Tc difference was 16.49 °C.

In the S1 model, Tb values tend to decrease until population percentage reaches 40%, where Tb was 9.02 °C; from this point on, the temperature tends to drop again until it reaches 80% (4.15 °C); To tends to increase until it reaches 40% and then decrease from this point until it reaches 60%, and then it rises from 60% to 80% (35.88 °C). Finally, for the same model, Tc tends to decrease from 10% to 30%; from this point on it was not possible to perform the respective linear regressions due to the lack of empirical data of 40–80%. For S2 Tb only tends to decrease from 9.62 to 2.59 °C; To only tends to increase; and Tb only tends to decrease; again, the lack of empirical data from 40–80% was a limitation to performing the corresponding regressions. For S3, Tb follows a tendency to decrease, To only increases, from 10% to 20% and subsequently remains constant near 30 °C, while Tc decreases from 10% to 30%, then increases at 40% and 50%, subsequently dropping from this point on. For all beta models, Tb tends to decrease, To remains constant in the three models, while Tc tends to increase in the B1 and B3 models, following an expected normal distribution, while in the B2 model Tc tends to decrease.

RMSE, R^2^, and adjusted R^2^ were calculated for all regression lines in the sub-optimal range in S1 and S2 models, while output parameters for 50–80% in the supra-optimal range of those models were unable to be calculated due to the lack of experimental data, where at least seven experimental points are needed to perform the corresponding regressions. For the remaining models, output parameters were calculated for a single regression that spanned both ranges. In all models the RMSE, R^2^, and adjusted R^2^ are prone to decrease as the population percentage increases; except for B3, where RMSE increases from 10% to 30% and from this point on tends to decrease. On the other hand, R^2^ and adjusted R^2^ increase from 10% to 20%, then decrease from 20% to 30%, afterward increasing from 30% to 50%, and finally decreasing from this point on.

## 3. Discussion

### 3.1. Germination

It has been reported that chia grows at a minimum temperature of 11 °C and a maximum of 36 °C, with an optimal between 16 and 26 °C [41]. However, until now, a wide germination temperature gradient had not been assayed. In this context, we observed that chia can germinate at lower temperatures, reaching a final germination percentage above 95% at 10 °C; at this temperature, germination is delayed because more time is needed to accumulate enough day degrees to complete germination, as has been observed in other short-day related species [39,42,43] and other phenological events of chia such as flowering [44]; however, up to today, a thermal time coefficient for chia germination is still lacking. The fact that germination is not completely inhibited at 10 °C indicates that the base temperature is below this value and lower concerning the previously reported temperature [41].

Our observed optimal condition, i.e., the temperature(s) at which the germination percentage is high, and germination occurs the fastest, in this case, 25–35 °C, is almost ten degrees above the reported range [41]; this evidence agrees with our previous results [24] and with those observed by Paiva et al. [25,26]. The differences in the germination of chia varieties generated by domestication are mainly associated with different capabilities to germinate and grow under different climatic conditions [27], like those varieties produced to grow during the long days of the northern hemisphere [45,46]. Another explanation is related to storage conditions, where it is already known that humidity and high temperatures reduce seed viability and germination [27]. These deterioration processes are related to the chemical composition of the seeds [47] and are mainly associated with factors such as water content, environmental conditions, microorganisms, and package and storage conditions, among others [48]. Because of the disruption of the membrane system caused by free radical attacks on the chemical components of the membrane, the most visible physiological symptoms of seed degeneration arise during germination and seedling initial development [49].

In other surveys carried out in some members of the *Salvia* genus, it has been observed that golden chia (*Salvia columbariae* Benth.) reached the highest final germination at 25 °C, compared with 4 °C and 10 °C [50], while Noroozak (*Salvia leriifolia* Benth.) has an optimal range between 15 and 25 °C, with a calculated Tb, To, and Tc of 1.00 °C, 19.0 °C, and 36.5 °C, respectively [51]. On the other hand, in an analysis conducted on 11 different medicinal plants [52], including two belonging to the *Salvia* genus, it was observed that *Salvia sclarea* L. and *Salvia nemorosa* L. reached the highest final germination and GR in the treatments at 20 °C for *S. sclarea* L. (final germination: 71.9%; GR: 65 seeds d^−^^1^; Tb: 0.0 °C; To: 21.0 °C and Tc: 40.3 °C) and 15 °C for *S. nemorosa* L. (final germination: 58.9%; GR: 54.3 seeds d^−^^1^; Tb: 0.0 °C; To: 17.0 °C and Tc: 41.0 °C).

The GR observed in our analysis was 1.4-fold and 2.2-fold higher than the one previously reported by Nadtochii et al. [53] for 25 °C, and 30 °C (13.1 ± 0.1 and 9.7 ± 0.1), while at 20 °C, the GR was similar between the works (12.6 ± 0.1 and 12.75 ± 0.72, respectively). In this subject, Adam et al. [54] showed that GR differed among species and seed lots within species. We observe that in chia, temperatures higher than 35 °C led to reduced final germination and germination rate, in this context it has been shown that temperature increased up to the optimum followed by increased GR, but declined afterward [40,55,56,57]. Hardegree [58] reported that there was a large error in predictions of seedling emergence in early spring due to seed degradation and lowering GR at high temperatures.

Our results suggest that *S. hispanica,* due to its oil quality and quantity, has an improved seed performance at low and higher temperatures, where a high proportion of polyunsaturated fatty acids helps to maintain cellular membrane fluidity; this agrees with our observation that chia, after a lag phase at 10 °C (four days) reaches final germination above 95%, in contrast with *S. sclarea* L. and *S. nemorosa* L., which reached less than 50% at temperatures below 10 °C and that, at a temperature above 40 °C, germination is significantly reduced, but not completely inhibited nonetheless [52], either by thermodormancy, thermoinhibition or directly by the death of the embryo [27,31,59]. This is supported by the fact that in *S. sclarea* L. and *S. nemorosa* L., seeds have been observed at lower concentrations of linoleic (both species with 15–20% less compared with *S. hispanica*) and α-linolenic acid (10% less in *S. sclarea* L. and almost 40% less in *S. nemorosa* L. compared with *S. hispanica*) [60,61].

Germination behavior is intrinsically connected to ecological conditions of species’ natural habitats and biogeographical origin [62]. Most of the species naturally found in arid and temperate climates have the potential to germinate well at temperatures ranging from 15 to 30 °C, indicating a preference for moderate and moderate-high temperatures for germination. In this subject, there is evidence that shows that temperatures ranging from 15 °C to 25 °C led to maximum germination in species characteristic of arid environments like *Lavandula dentata* L., *Teucrium gnaphalodes* L’Hér., *Thymbra capitata* (L.) Cav., and *Thymus hyemalis* L., while higher temperatures limit germination and growth [63]. Therefore, it is not surprising that temperatures < 10 °C and > 35 °C affect the growth and yield of chia, either delaying or inhibiting germination, considering that this species is adapted to Central and South American annual mean temperatures that fluctuate between 11 and 36 °C [17,18,19,41], reflecting their distribution and climatic conditions for optimal germination.

### 3.2. Cardinal Temperature Determination by Linear and Non-Linear Regression Models

The temperature variation range for all models was quite different, while with the S1 and S2 models it was not possible to determine Tc for 4 percentiles (40–80%), due to restrictions related to the number of experimental points needed to perform the regression analysis. In all segmented models, the Tb values were less variable than To and Tc values. Finally, Tc variations were narrower for S1 and S2 models, considering that only 3 percentiles were able to estimate with these models. On the other hand, for beta models Tb variation was wider between percentiles than To and Tc variation, with To variation being the narrowest of all beta models; the B3 model estimated widely for Tb and Tc variation range.

Our observed Tb downward trends in S1–S3 were according to those tendencies reported for Tb determination with two and three-piece segmented models observed in *Phalaris minor* Retz. [64] and *Silybum marianum* L. [65]. The observed trends in this species include a linear increase in germination, followed by a plateau segment, and finally a decrease in germination until reaching zero as the temperature increases; the plateau segment was not observed in our data of chia germination. Our data tendency involves higher slope values for lower population percentages, positioning regression lines from higher to lower values of x-intercept (Tb values) as the percentage of the population increases. The same phenomenon seems to occur with the Tb estimation with B2 and B3 models, whose observed trends are opposite to the observed Tb values for beta functions with five-parameters [64,65], while our observed tendency for B1 (beta four parameters or modified) is in accordance with the one observed in *S. marianum* L. [65]. The Tb tendencies for B2 and B3 models are related to chia 1/VG data distribution. While the physiological meaning of this behavior remains to be explored, it has been suggested that the amount of energy reserve of the different population percentages could be related to a higher germination efficiency, and related in turn with faster germination, where more energy is needed to satisfy the metabolic demands of this population, especially for processes such as those of starch and protein hydrolysis [66].

Our findings established that the optimal temperature range of chia seed germination was from 28.4 to 36.97 °C, with a maximum GR observed at 33.45 ± 2.76 °C in S1, 36.97 ± 5.70 °C in S2, 32. 60 ± 1.20 °C in S3, 30.45 ± 0.32 °C in B1, 28.44 ± 2.28 °C in B2, and 30.34 ± 1.11 °C in B3; these values did not support previous findings, where an optimal range between 16 and 26 °C has been observed [17,18,19,41]. Until now, there are few reports where the cardinal temperatures of some species of the genus *Salvia* have been determined. In this context, it has been reported that seeds of *Salvia pomifera* L. and *Salvia fruticosa* Mill. have an optimal temperature range of 10–20 °C [67]. Moreover, it has been found that seeds of *Salvia officinalis* L. germinated within the range of 10–25 °C and that *Salvia sclarea* L. had a broader range of optimal temperatures from 10 to 30 °C [68], while in another survey it was reported that the optimum temperature for seed germination of *S. officinalis* L. was 25 °C [69]. For the majority of plant species, optimum and ceiling temperatures have been reported at 15–30°C and 30–40°C, respectively [69]; however, the optimum temperature of germination depends on genetic and environmental conditions that the plant evolved [70].

Finally, it has been reported that cardinal temperatures for *Plantago ovata* Forssk. were Tb: 4.4 °C, To: 19.0 °C, and Tc: 25.5 °C, while for *Plantago psyllium* L. they were Tb: 9.4 °C, To: 28.8 °C, and Tc: 35.0 °C [56]; it is important to emphasize that these two species are myxospermic angiosperms with copious mucilage as chia, and that they also belong to the Lamiales order. Mucilage is a polymer secreted by a variety of plants and their parts, including *Aloe vera* L., *Salvia hispanica* L. seeds, *Cordia dichotoma* G.Forst., *Basella alba* L., *Plantago psyllium* L., *Cyamopsis*
*tetragonoloba* (L.) Taub., *Cactaceae*, *Abelmoschus esculentus* (L.) Moench, *Trigonella foenum-graecum* L., *Moringa oleifera* Lam., and *Linum usitatissimum* L. [71]. The structure, components, ecological roles, and production mechanism of mucilage have been well studied in the model plant *Arabidopsis thaliana* (L.) Heynh. [72,73,74]. Studies involving *A. thaliana* (L.) Heynh. and other species have concluded that seed coat mucilages may have multiple roles, including the inhibition of germination under excessive moist conditions (i.e., seed dormancy) by preventing the embryo from oxygen diffusion [74]. Evidence with the seed mucilage *Lavandula subnuda* Benth. (Lamiaceae) and *Plantago ciliata* Desf. (Plantaginaceae) has established that mucilage presence increased moisture uptake and inhibited germination at lower temperatures [74]. Upon germination, the progressive depletion of oxygen generates conditions that almost achieve anaerobiosis, and fermentation is triggered as the main source of cellular ATP, supporting the reduction of electron transferring compounds, e.g., NAD and NADP, and inevitably leading to ROS (reactive oxygen species) accumulation [75]. In the case of *S. hispanica*, we observed that in a temperature range from 10 °C to 20 °C, germination undergoes a lag phase; above that temperature, germination increases exponentially as it approaches the optimal temperature, this evidence suggests that mucilage could have a temperature where its moisture-holding capacity changes, allowing germination to proceed more quickly; this agrees with our previous observations [24].

Parameters from model fitting to the reciprocal of GR versus temperature data are shown in Table 3, Table 4 and Table 5 for S1–S3 models and Table 6 for B1–B3 models. For the sub- and supra-optimal range in S1, the root means square of deviations (RMSE) was highest for 10% and tends to decrease as the percentage of the population increases in the sub-optimal range (RMSE from 1.28 to 0.11); the same tendency was observed from 10% to 30% in the supra-optimal range (RMSE from 1.88 to 0.55 for the supra-optimal range). The same tendency was observed for S2 and S3 models with values from 1.46 to 0.11 and 1.88 to 0.11, respectively. R^2^ and adjusted R^2^ were >0.7 for 10–70 percentiles of all segmented models, except for the lowest values of the adjusted R^2^ and adjusted R^2^ of 80% (<0.6) in the S1–S3 models. RMSE was lower for beta models than segmented ones, except for the B2 model ranging from 1.53 to 0.13 for the B1 model, from 2.26 to 0.22 for B2, and from 1.66 to 0.03 for the B3 model; these lower RMSE values of B1 and B3 indicate a higher fit of the beta models to our empirical data, with globally B3 having the best fitting output parameters; however, this model tends to overestimate Tb values (12.74 ± 4.45 °C).

While all six models showed a good predicting ability, beta models had a better estimate for cardinal temperatures. For all S models a better fit can be observed as the population percentage increases; this type of function has been used for the description of data distribution with little variation in the germination rate between percentiles in an optimal range, forming a plateau, as those observed in chickpea [76], littleseed canarygrass [64], and milk thistle [65]; this behavior is in contrast to the type of performance that we observed in chia, where three abrupt changes in the germination rate were observed at 20 °C, 30 °C, and 40 °C, respectively. Moreover, it has been observed that segmented models tend to overestimate base or maximum temperature when only two segments are used, making a bilinear function [77]; in the case of chia, however, we observed a realistic value for all models, including segmented ones, where the only exception was observed in the B3 model (Tb of 12.74 ± 4.45 °C), while To and Tc were quite similar for all models. This evidence agrees well with other findings where it has been observed that segmented functions adequately described the response of germination, leaf appearance, and development rate to temperature in different crops [78,79,80].

On the other hand, the observed lower values of RMSE and higher R^2^ and adjusted R^2^ for beta models are expected because beta functions are more flexible than non-linear functions with segments, due to the curvilinear nature of beta models that provide a gradual transition between phases producing a smooth realistic curve. Beta models did not require the determination of cardinal temperatures for each subpopulation and therefore the models can be easily parameterized since they can be linearized if values of Tb and T are predetermined from the data or external sources, or data transformation to probit units are not required [58,81]. Moreover, it has been observed that curvilinear models accurately predict ceiling temperatures by extrapolation when empirical data are not available [82]. Beta functions show some limitations, i.e., they assume a symmetric response about optimum temperature and do not allow for any concave curvature near the base temperature [81]; however, exponential functions, like those used for B3, are flexible enough to handle nonsymmetric responses; these characteristics could explain the overestimation of the base temperature by this model, making it accurate and suitable for chia To and Tc but not for Tb.

The results of this work indicate that all assayed models fit empirical data of chia germination well in response to temperature; however, beta models have a better fit than segmented ones and that the B1 model sharply defines the cardinal temperatures of *S. hispanica*.

## 4. Materials and Methods

### 4.1. Seed Acquisition and Store

Medicinal variety *S. hispanica* seeds were obtained without previous treatment and with 90% germination and 99% purity in accordance with the supplier (Okko super foods^©^; Jalisco, México; Lot/Batch: 130320/19). Seeds were kept in their shipping bags in a cold, dry seed storeroom at 10 ± 5 °C and 20 ± 5% relative humidity (RH) until imbibition assays were conducted. No previous disinfection treatment was applied in any of the experiments due to chia seeds’ response at mucilage secretion level [83,84].

### 4.2. Germination Tests

Five replicates of 25 seeds were sown randomly on an agar medium (10 g L^−1^) in Petri dishes (5.5 × 1.5 cm). Seeds were incubated at constant temperatures in germination chambers at 10 °C to 45 ± 2 °C (with intervals of 5 °C) with a 12 h photoperiod, the same as Cabrera-Santos et al. [24] and Sampayo-Maldonado et al. [85]. Seeds were considered germinated when the radicle emerged ≥ 2 mm [86]; after that, seedlings were removed from the Petri dish. Germination was recorded daily for 14 days, a time at which no more germination was observed.

### 4.3. Variables Evaluated

#### 4.3.1. Total Germination

The daily number of germinated seeds in each Petri dish was recorded. G (%) was reported as the average cumulative percentage of germinated seeds in each treatment, calculated according to:(1)$$G\;\left(\%\right)=\frac{n}{N}\times 100$$
where *n* is the number of seeds germinated and *n* is the total number of seeds.

#### 4.3.2. Median Germination Time (t_50_)

The total number of days between imbibition time and when 50% of the total germination was recorded. According to Ordoñez-Salanueva et al. [87], a sigmoid curve was fitted to the accumulated germination, allowing the median germination time to be determined by interpolation.

#### 4.3.3. Germination Rate (GR)

Germination rate or the number of germinated seeds by day was obtained with the equation proposed by Maguire [88]:(2)$$\mathrm{GR}=\frac{G_{1}}{N_{1}}+\frac{G_{2}}{N_{2}}+\ldots +\frac{G_{i}}{N_{i}}+\frac{G_{n}}{N_{n}}=\overset{n}{\underset{i=1}{\sum }}\frac{G_{i}}{N_{i}}\;$$
where *G_i_* is the number of germinated seeds and *N_i_* is the number of days after the beginning of the experiment.

The reciprocal of time for germination for each fraction of the seed population (10–100%) was calculated and plotted as a function of the temperature to observe the tendency of the data, locate the point of inflection, and determine the sub-optimal and supra-optimal temperature ranges.

### 4.4. Determination of Cardinal Temperatures by Non-Linear and Linear Regression Models

To formulate and validate mathematical functions that have been used to quantify the effect of temperature on the biological time required for germination and cardinal temperature determination, the reciprocal of the germination data at sub-optimal and supra-optimal temperatures was used to perform the different models; mathematical modeling was based on the available literature [64,65,89,90,91,92].

Segmented models 1 and 2 (S1 and S2, respectively) included a pair of two-segment linear regressions (one at sub-optimal and the other at the supra-optimal temperature range); the optimum temperature was the temperature at which these two lines intersect [58]. The first segment of the regression in the sub-optimal range was used to estimate the x-intercept of each regression line, i.e., base temperature or Tb, while for the supra-optimal range the second segment was used to estimate the x-intercept, i.e., ceiling temperature or Tc [27]. An average of the x-intercept among fractions in the sub-optimal and supra-optimal temperature range was calculated to establish the Tb and Tc [93]. Parameters for 10–40% only were obtained with these models, due to the few available points of 50–80% percentiles at supra-optimal range to perform the respective regressions (at least five experimental data points).

For models S1 and S2, the two segments of the linear regression in the suboptimal range were constructed with the condition *X* < *X*0 for the first segment and *X* > *X*0 for the second segment, where: *X* = any value of *X* and *X*0 = value of the *X* coordinate where the two segments meet. For the S1 model, linear regressions in both temperature ranges were performed letting *X*0 vary without any restriction. On the other hand, abrupt breakpoints (statistically different) in slope value were observed in both ranges at 20 °C and 40 °C, for sub- and supra-optimal ranges, respectively; therefore, the S2 model was restricted with *X*0 at those temperatures.

S3 was constructed with a three segment linear regression following the criteria *X* < *X*0 for the first and the second segments, while *X* < *X*1 was used for the third segment. For this model, *X*0 and *X*1 were allowed to vary without restrictions.

Beta models were based on beta probability density distribution, often used for fitting curvilinear relationships. B1 and B2 models were performed varying α and β parameters (α = 5, β = 4 for B1 and α = 8, β = 6 for B2, respectively) without any other restriction. B3 models were performed with the function reported in Reyes-Ortega [94].

To determine the best estimates of the parameters (lower deviations of the intercept from 0 and of the slope from 1 correspond to increased reliability (RMSE; Equation (3)), the coefficient of determination (R^2^; Equation (4)), and the intercept and slope of the regression of predicted vs. observed germination rate were used.
(3)$$\mathrm{RMSE}=\sqrt{\left(\frac{1}{n}\right)\sum^{\;}{\left(Y_{obs}-Y_{pred}\right)}^{2}}$$
where *Y_obs_* denotes observed value, *Y_pred_* predicted value, and *n* the number of samples [95], and
(4)$$R^{2}=SSR/SST$$
where *SSR* denotes the sum of squares (*SS*) for regression$\;\left(\overset{n}{\underset{i=1}{\sum }}\left(\hat{Y}-\bar{Y}\right)\right)$ and *SST* the total *SS* $\left(\overset{n}{\underset{i=1}{\sum }}\left(Y_{i}-\bar{Y}\right)\right)$. *Yi* is the observed value and *Y* is the correspondent estimated value. Low RMSE and R^2^ near 1 correspond to better model estimation.

Segmented and beta models were constructed with non-linear regressions with two or three segments based on Soltani et al. [76] for S3 model, Yin et al. [81] for B1 and B2 models, and Reyes-Ortega [94] for B3 model (Table 2), respectively. TableCurve^®^ 2D (version 5.01 for windows, Systat Software Inc., San Jose, CA, USA, www.sigmaplot.co.uk/products/tablecurve2d, accessed on 15 September 2021) and GraphPad Prism^®^ software (version 8.4.0 for macOS, GraphPad Software, San Diego, CA, USA, www.graphpad.com, accessed on 25 September 2021) were used to calibrate the models via the iterative least square method.

### 4.5. Statistical Analysis

Germination data did not fulfill the assumption of normality either with transformed (Arcsine, Logit and Ln) or non-transformed data; therefore, significant differences in total germination, median germination time (t_50_), and germination rate (GR) were determined by Kruskal–Wallis followed by Dunn’s post hoc test for multiple comparisons. Statistical analyses were carried out using the GraphPad Prism^®^ software, version 8.4.0 for macOS, GraphPad Software, San Diego, CA, USA, www.graphpad.com (accessed on 10 October 2021).

## 5. Conclusions

In this work, we explore the effect of temperature on seed germination in the oilseed crop *S. hispanica*. The main conclusions are the following:Germination of *S. hispanica* L. occurs in cold to moderate high temperatures (10–30 °C), having an optimal range between 25 and 35 °C, with the highest GR and t_50_ observed at 30 °C. The temperatures higher than 35 °C strongly inhibited the germination characteristics.The results of this study showed that the response of chia germination to temperature was best explained by beta models, having a better fit than segmented models.Cardinal temperatures for chia germination calculated by the B1 model were: 2.52 ± 6.82 °C for the base, 30.45 ± 0.32 °C for optimum, and 48.58 ± 2.93 °C for ceiling temperature.

This is the first report of cardinal temperature determination of *Salvia hispanica* L.; our data for the base, optimum, and ceiling temperatures for chia seed germination provide basic temperature requirements that can be used in further research and cropping of this species; further assays must be oriented to determine thermal requirements of the different chia genotypes and varieties. As a perspective, it is necessary to carry out approaches in -omics fashion (genomic, proteomic, and metabolomic), in order to have a more complete physiological overview; meanwhile, every effort must be oriented towards its application in the field, which by virtue of the economic and ecological situations that our societies are going through, attends to an activity of primary importance: food.

## Figures and Tables

**Figure 1 plants-11-01142-f001:**
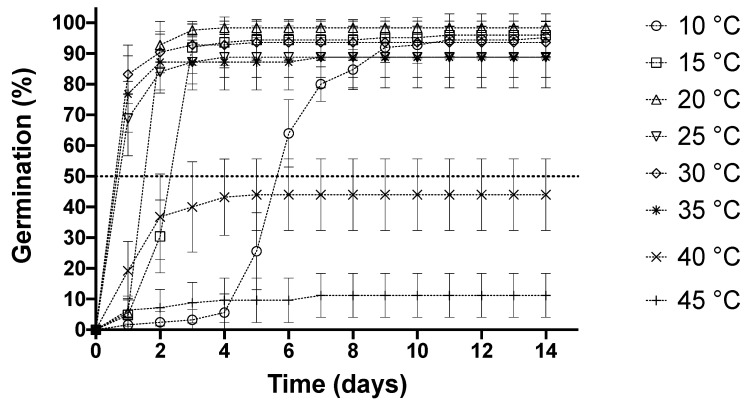
Cumulative germination of *S. hispanica* at 10, 15, 20, 25, 30, 35, 40, and 45 °C. Values are expressed as mean ± SD of five independent replicates. Statistical analysis was performed using Kruskal-Wallis followed by a Dunn’s multiple comparison test.

**Figure 2 plants-11-01142-f002:**
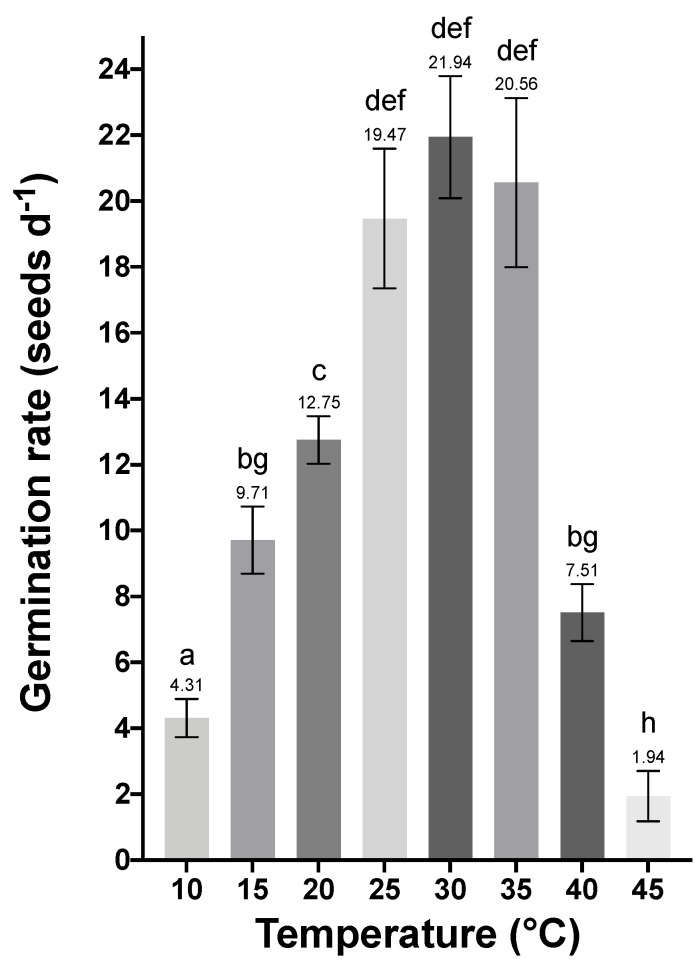
Germination rate per day of seeds at 10–45 °C. Values are shown as means ± SD of five replicates with 25 seeds each. Values are expressed as mean ± SD of five independent replicates. Statistical analysis was performed using Kruskal-Wallis followed by a Dunn’s multiple comparison test. Different letters indicate significant differences.

**Figure 3 plants-11-01142-f003:**
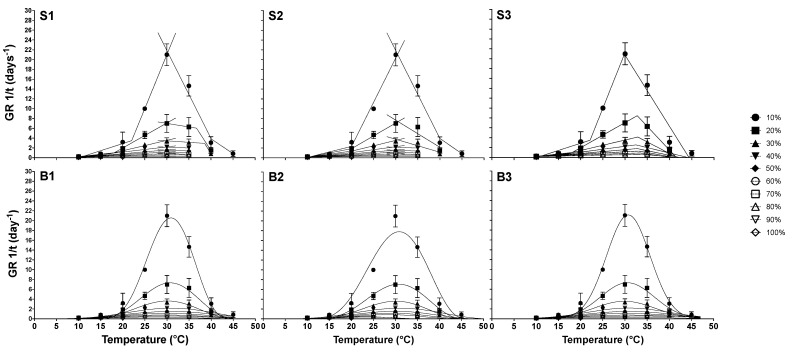
Relationship between the reciprocal of the GR and the germination temperature of the percentiles in the cardinal temperature range. Symbols represent experimental data, while the solid lines correspond to the predicted values by segmented (**S1**–**S3**) and beta functions (**B1**–**B3**).

**Table 1 plants-11-01142-t001:** Final germination percent and median germination time (t_50_) of seeds during imbibition at 10, 15, 20, 25, 30, 35, 40, and 45 °C. Final germination is the percentage of seeds in which the germination process reaches the end; while median germination time (t_50_) is the time to reach 50% of final germination.

Temperature	Final Germination (%)	Median Germination Time t_50_ (Days)
10 °C	95.20 ± 5.21 ^a^	5.50 ± 0.44 ^a^
15 °C	96.00 ± 6.92 ^a^	2.00 ± 0.15 ^b^
20 °C	98.40 ± 2.19 ^a^	1.32 ± 0.07 ^c^
25 °C	88.80 ± 6.57 ^a^	0.56 ± 0.19 ^d^
30 °C	93.60 ± 5.36 ^a^	0.30 ± 0.10 ^d^
35 °C	88.80 ± 9.96 ^a^	0.38 ± 0.22 ^d^
40 °C	44.00 ± 11.66 ^b^	ND
45 °C	11.20 ± 7.15 ^c^	ND

Values are expressed as mean ± SD of five independent replicates. Statistical analysis was performed using Kruskal-Wallis followed by a Dunn’s test multiple comparison test. The values that share the same letters did not present statistically significant differences.

**Table 2 plants-11-01142-t002:** Non-linear regression models fitted to reciprocal of GR versus temperature data for 10–80% percentiles to determine cardinal temperatures of chia seeds.

Model; Reference	Formula	Conditions	Description
Segmented 1; S1	Y1=b1+m1∗X	First segment (*Y*1): *X* < *X*0 for sub- and supra-optimal ranges	Two-segments no-linear regressions model with intersected lines for 10–80% of the seed population. *X*0 no constricted
	YatX0=m1∗X0+b1	*X* coordinate where the two segments meet (*YatX*0)	
	Y2=YatX0+b2∗(X−X0)	Second segment (*Y*2): *X* > *X*0 for sub- and supra-optimal ranges	
Segmented 2; S2	Y1=b1+m1∗X	First segment (*Y1*): X < X0 for sub- and supra-optimal ranges	Two-segments no-linear regressions model with intersected lines for 10–80% of the seed population. Sub-optimal *X*0 constricted at 20 °C; supra-optimal *X*0 constricted at 40 °C.
	YatX0=m1∗X0+b1	X coordinate where the two segments meet (*YatX*0)	
	Y2=YatX0+b2∗(X−X0)	Second segment (*Y*2): *X* > *X*0 for sub- and supra-optimal ranges	
Segmented 3; S3	Y1=b1+m1∗X	First segment (*Y*1): *X* < *X*0 for sub- and supra-optimal ranges	Three-segment non-linear regression; no-constrained.
	YatX0=m1∗X0+b1	*X* coordinate where *Y1* and *Y*2 segments meet (*YatX*0)	
	Y2=YatX0+b2∗(X−X0)	Second segment (*Y*2): *X* > *X*0 and *X* < *X*1 for sub- and supra-optimal ranges	
	YatX1=YatX0+(X1−X0)∗ m2	*X* coordinate where *Y*1 and *Y3* segments meet (*YatX*1 or *To*)	
	Y3=YatX1+m2∗(X−X1)	Third segment (*Y*3): *X* > *X*1 for sub- and supra-optimal ranges	
Beta 1; B1	$f\left(T\right)={\left(\frac{T-Tb}{To-Tb}\right)}^{\alpha }\times {\left(\frac{Tc-T}{Tc-To}\right)}^{\beta }$	α = 5; β = 4	Four-parameters; One non-linear regression; no-constrained.
Beta 2; B2	$f\left(T\right)={\left({\left(\frac{T-Tb}{To-Tb}\right)}^{\frac{Tc-To}{To-Tb}}\right)}^{\alpha }\times {\left({\left(\frac{Tc-T}{Tc-To}\right)}^{\frac{Tc-To}{To-Tb}}\right)}^{\beta }$	α= 8; β = 6	Five-parameters; One non-linear regression; no-constrained.
Beta 3; B3	$f\left(T\right)=\left(A0\right)\times (e^{(-A1\times {\left(X/A2-1\right)}^{2+1/\left(X-A3\right)}}$)		One non-linear regression; no-constrained.

**Table 3 plants-11-01142-t003:** Estimated parameters for segmented model (S1) of *Salvia hispanica* seeds. Root mean square of deviations (RMSE) and coefficient of determination (R^2^) for the relationship between emergence rates.

Parameter	S1															
	10%		20%		30%		40%		50%		60%		70%		80%	
**Tb (°C)**	10.09		7.45		7.29		9.02		6.86		6.56		4.93		4.15	
**Mean Tb (°C)**	**6.90 ± 1.86**															
**To (°C)**	31.09		31.43		32.54		37.49		36.28		29.93		32.98		35.88	
**Mean To (°C)**	**33.45 ± 2.76**															
**Tc (°C)**	47.31		40.77		40.40		ND		ND		ND		ND		ND	
**Mean Tc (°C)**	**42.83 ± 3.88**															
**Range**	**Sub**	**Supra**	**Sub**	**Supra**	**Sub**	**Supra**	**Sub**	**Supra**	**Sub**	**Supra**	**Sub**	**Supra**	**Sub**	**Supra**	**Sub**	**Supra**
**RMSE**	1.28	1.88	0.91	1.46	0.44	0.55	0.32	ND	0.22	ND	0.17	ND	0.13	ND	0.11	ND
**R^2^**	0.97	0.95	0.88	0.75	0.88	0.83	0.84	ND	0.83	ND	0.80	ND	0.74	ND	0.58	ND
**Adjusted R^2^**	0.97	0.94	0.87	0.68	0.87	0.78	0.82	ND	0.81	ND	0.77	ND	0.71	ND	0.52	ND

**Table 4 plants-11-01142-t004:** Estimated parameters for the segmented model (S2) of *Salvia hispanica* seeds. Root mean square of deviations (RMSE) and coefficient of determination (R^2^) for the relationship between emergence rates.

Parameter	S2															
	10%		20%		30%		40%		50%		60%		70%		80%	
**Tb (°C)**	9.62		9.41		8.74		8.05		6.96		6.10		4.67		2.59	
**Mean Tb (°C)**	**6.65 ± 2.55**															
**To (°C)**	31.07		32.28		33.29		36.05		36.70		37.71		39.51		49.22	
**Mean To (°C)**	**36.97 ± 5.70**															
**Tc (°C)**	46.64		44.06		44.19		ND		ND		ND		ND		ND	
**Mean Tc (°C)**	**44.96 ± 1.45**															
**Range**	**Sub**	**Supra**	**Sub**	**Supra**	**Sub**	**Supra**	**Sub**	**Supra**	**Sub**	**Supra**	**Sub**	**Supra**	**Sub**	**Supra**	**Sub**	**Supra**
**RMSE**	1.46	1.88	0.93	1.76	0.45	0.74	0.31	ND	0.22	ND	0.17	ND	0.13	ND	0.11	ND
**R^2^**	0.96	0.95	0.88	0.63	0.88	0.69	0.85	ND	0.83	ND	0.80	ND	0.74	ND	0.58	ND
**Adjusted R^2^**	0.96	0.94	0.87	0.57	0.87	0.64	0.83	ND	0.82	ND	0.78	ND	0.72	ND	0.54	ND

**Table 5 plants-11-01142-t005:** Estimated parameters for segmented models (S3) of *Salvia hispanica* seeds. Root mean square of deviations (RMSE) and coefficient of determination (R^2^) for the relationship between emergence rates.

Parameter	S3							
	10%	20%	30%	40%	50%	60%	70%	80%
**Tb** (**°C**)	10.09	7.45	7.29	7.09	6.86	6.50	5.91	0.99
**Mean Tb** (**°C**)	**6.52 ± 2.55**							
**To** (**°C**)	29.84	32.73	32.85	32.43	32.72	33.04	33.37	33.89
**Mean To** (**°C**)	**32.60 ± 1.20**							
**Tc** (**°C**)	44.46	41.05	39.85	45.71	44.73	36.59	37.00	32.84
**Mean Tc** (**°C**)	**41.34 ± 3.74**							
**RMSE**	1.88	1.04	0.47	0.32	0.22	0.18	0.14	0.11
**R^2^**	0.94	0.86	0.87	0.85	0.83	0.79	0.70	0.64
**Adjusted R^2^**	0.93	0.84	0.85	0.81	0.80	0.74	0.64	0.56

**Table 6 plants-11-01142-t006:** Estimated parameters for beta models (B1–B3) of *Salvia hispanica* seeds. Root mean square of deviations (RMSE) and coefficient of determination (R^2^) for the relationship between emergence rates.

Parameter	B1 (Four-Parameters)								B2 (Five-Parameters)								B3							
	10%	20%	30%	40%	50%	60%	70%	80%	10%	20%	30%	40%	50%	60%	70%	80%	10%	20%	30%	40%	50%	60%	70%	80%
**Tb** (**°C**)	12.88	9.38	6.85	4.42	−1.22	−2.06	−5.57	−4.50	9.97	9.98	8.14	6.19	0.71	0.49	2.18	2.16	20.10	17.11	15.37	12.91	10.33	10.05	8.80	7.25
**Mean Tb** (**°C**)	**2.52 ± 6.82**								**4.97 ± 4.06**								**12.74 ± 4.45**							
**To** (**°C**)	31.03	30.86	30.24	30.47	30.24	30.22	30.39	30.16	30.97	30.92	30.19	29.81	27.31	27.13	25.68	25.55	30.19	32.78	30.15	30.30	30.79	29.94	29.54	29.06
**Mean To** (**°C**)	**30.45 ± 0.32**								**28.44 ± 2.28**								**30.34 ± 1.11**							
**Tc** (**°C**)	44.28	46.31	46.44	46.93	51.32	50.28	52.33	50.75	46.48	46.06	46.26	47.04	44.45	43.5	41.26	39.1	40.22	41.34	44.65	49.92	55.85	50.06	53.14	56.71
**Mean Tc** (**°C**)	**48.58 ± 2.93**								**44.26 ± 2.83**								**48.99 ± 6.32**							
**RMSE**	1.53	1.10	0.53	0.36	0.23	0.19	0.15	0.13	2.26	1.10	0.52	0.36	0.36	0.28	0.25	0.22	1.66	1.72	2.30	0.31	0.22	0.18	0.15	0.03
**R^2^**	0.96	0.85	0.84	0.81	0.82	0.77	0.68	0.51	0.91	0.85	0.84	0.81	0.55	0.47	0.11	−0.37	0.99	0.96	0.95	0.99	0.99	0.98	0.98	0.96
**Adjusted R^2^**	0.96	0.84	0.83	0.80	0.81	0.75	0.65	0.47	0.90	0.84	0.83	0.80	0.52	0.43	0.04	−0.48	0.98	0.93	0.90	0.97	0.98	0.96	0.95	0.92

## Data Availability

The data presented in this study are available on request to the corresponding author.

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
