# Peer review of "Quantifying Cardinal Temperatures of Chia (Salvia hispanica L.) Using Non-Linear Regression Models"

_plants, 2022, doi:10.3390/plants11091142_

Round 1

Reviewer 1 Report

Please find my comments in the file attached.

Reviewer 2 Report

Cabrera-Santos et al presented a model for the estimation of cardinal temperatures of chia. The article is within the scope of Plants and can be accepted for publication. My only concern is the usage of only one variety, thus the absence of the evaluation of the effect of the genotype.  

Author Response

The suggestion was taken into account and line 543 was added to refer the need for further experiments comparing more genotypes.

Reviewer 3 Report

As a statistician, I have mostly focussed on the modeling and statistical methods decribed in this paper. In general I found the description of the methods very confusing. That means that (a) it is very hard to understand what has been done and thus whether it has been done correctly, and (b) it would be really hard to replicate this analysis even for a trained statistician, not to mention a statistical practicioner. The fact that the authors used menu-based rather than code-based software compounds the problem of replicability.

My comments below are based on what I could understand, but the methods need to be rewritten in a clearer way before a proper judgement can be made.

The first paragraph on p.3 states that no significant differences were observed in final germination percentage between the treatments. It then proceeds to report an H-statistic, and the p-values for Kruskall-Wallis and Dunn's test. Kruskall-Wallis test is a non-parametric test based on rank and actually compares distributions rather than their means. It is probably better to do a parametric anova on (logit-transformed) proportions, rather than the Kruskall-Wallis test. Furthermore, the very first sentence says that no significant differences were observed, but then mentions p<0.001 for Kruskall-Wallis, which is contradictory. The post-hoc test is usually used to figure out which specific pairs of groups are different from each other. If that is not of interest, then why do it at all? If it is, then which specific pairwise comparison does the single p-value refer to?

I found the description of the model fitting in section 4.4 extremely confusing and hard to follow. The authors mention "combination" of models and "intersection" of models. I don't think those are standard terms. I am not sure what the adjustment of linear regression in lines 474-485 refers to. It is also unclear to me what is meant by, for example, the condition if T_b<T<=T_0 for 10-30C in Table 2, S1. What does the 10-30 range refer to? In S2 it gets even more complicated: if T_b<T<=T_0 for 10-20C & 20-30C.

On the whole it reads like a very complex multi-stage ad-lib tweaking, which is never a good fitting procedure. Any uncertainty measures (such as confidence intervals) are automatically produced for the finally fitted model. They do not take into account the fact that the final model is a result of a step-wise procedure. That makes them meaningless in this context. The standard method of fitting segmented regression treats the change points (Tb, Tc, and To in this case) as model parameters and fits them along with other model parameters. 

As for the beta models, why were the specific parameter values (alpha=5, beta=4 and alpha=8, beta=6) chosen? Again, potentially, those can be estimated during model fitting directly rather than forced exogeneously. 

Author Response

See the attached Point to Point file
